# A Novel Anti-PD-L1 Vaccine for Cancer Immunotherapy and Immunoprevention

**DOI:** 10.3390/cancers11121909

**Published:** 2019-12-01

**Authors:** Jie Chen, Hui Liu, Tiffany Jehng, Yanqing Li, Zhoushi Chen, Kuan-Der Lee, Hsieh-Tsung Shen, Lindsey Jones, Xue F. Huang, Si-Yi Chen

**Affiliations:** 1Pomona Biotechnology Corp., 605 E Huntington Drive, Monrovia, CA 91016, USA13811984577@139.com (Y.L.); czs850723@163.com (Z.C.); 2Department of Molecular Microbiology and Immunology, Norris Comprehensive Cancer Center, Keck School of Medicine, University of Southern California, 1450 Biggy St, Los Angeles, CA 90033, USA; liuhui806@gzhmu.edu.cn (H.L.); jehng@usc.edu (T.J.); lindseyawoodham@gmail.com (L.J.); 3Department of Hematology and Oncology, Taipei Medical University Hospital and Department of Medicine, Taipei Medical University, Taipei 110, Taiwan; cj69888@163.com (K.-D.L.); d622107003@tmu.edu.tw (H.-T.S.); 4Jiyan Biomedical Corp., Taipei 110, Taiwan

**Keywords:** PD-L1, dendritic cells, tumor vaccine, immune checkpoint, immunotherapy

## Abstract

Dendritic cells (DCs) are potent antigen-presenting cells that play a critical role in activating cellular and humoral immune responses. DC-based tumor vaccines targeting tumor-associated antigens (TAAs) have been extensively tested and demonstrated to be safe and potent in inducing anti-TAA immune responses in cancer patients. Sipuleucel-T (Provenge), a cancer vaccine of autologous DCs loaded with TAA, was approved by the United States Food and Drug Administration (FDA) for the treatment of castration-resistant prostate cancer. Sipuleucel-T prolongs patient survival, but has little or no effect on clinical disease progression or biomarker kinetics. Due to the overall limited clinical efficacy of tumor vaccines, there is a need to enhance their potency. PD-L1 is a key immune checkpoint molecule and is frequently overexpressed on tumor cells to evade antitumor immune destruction. Repeated administrations of PD-L1 or PD-1 antibodies have induced sustained tumor regression in a fraction of cancer patients. In this study, we tested whether vaccinations with DCs, loaded with a PD-L1 immunogen (PDL1-Vax), are able to induce anti-PD-L1 immune responses. We found that DCs loaded with PDL1-Vax induced anti-PD-L1 antibody and T cell responses in immunized mice and that PD-L1-specific CTLs had cytolytic activities against PD-L1^+^ tumor cells. We demonstrated that vaccination with PDL1-Vax DCs potently inhibited the growth of PD-L1^+^ tumor cells. In summary, this study demonstrates for the first time the principle and feasibility of DC vaccination (PDL1-Vax) to actively induce anti-PD-L1 antibody and T cell responses capable of inhibiting PD-L1^+^ tumor growth. This novel anti-PD-L1 vaccination strategy could be used for cancer treatment and prevention.

## 1. Introduction

Dendritic cells (DCs) are potent antigen-presenting cells and play a critical role in activating cellular and humoral immune responses [1,2,3,4,5,6,7,8,9]. Sipuleucel-T (Provenge), a cancer vaccine of autologous DCs, loaded with TAA, was approved by the United States Food and Drug Administration for the treatment of castration-resistant prostate cancer. Sipuleucel-T prolongs patient survival, but has little or no effect on clinical disease progression or biomarker kinetics. Liau LM et al. recently reported that vaccinations with an autologous tumor lysate-pulsed dendritic cell (DCVax-L) after surgery and chemoradiotherapy likely extended survival in patients with newly diagnosed glioblastoma in a Phase Three trial [10]. Recently, we found that a genetically modified DC vaccine elicited potent tumor-associated antigen-specific T cell responses and yielded improved survival rate in acute leukemia patients [11]. We further observed a complete remission rate of 83% in 12 relapsed acute myeloid leukemia (AML) patients [11]. However, so far, their clinical benefit has been limited [12,13,14,15,16,17,18]. Evidently, in the face of activated antitumor T cell responses, there are immune suppressive mechanisms that allow tumors to escape immune destruction [19,20,21,22,23], underlining the need to develop strategies to counter these immune suppressive mechanisms.

Tumor cell expression of Programmed Death Ligand 1 (PD-L1, B7-H1) represents a major immune checkpoint for suppressing antitumor T cell responses [21,22,23]. Many tumors, including lung, ovarian, melanoma, and pancreatic tumors, express PD-L1 [21,22,23,24]. PD-L1 is also expressed in tumor-associated myeloid-derived suppressor cells (MDSCs) in the tumor microenvironment [25,26,27,28]. Ligation of PD-L1 with its binding partner PD-1 results in the downregulation of proliferative responses, decreased cytokine secretion, and T cell anergy or apoptosis [21,22,24]. Clinical trials show that repeated administrations of anti–PD-1 or anti–PD-L1 antibodies produce durable responses in a subset of cancer patients by releasing the effector functions of naturally occurring antitumor cytotoxic T lymphocytes (CTLs) [21,22,29,30,31,32]. However, a significant portion of cancer patients did not respond to these immune checkpoint inhibitors, which may be limited by the absence of sufficient antitumor T cells in these cancer patients.

In this study, we tested whether vaccinations with DCs loaded with a PD-L1 immunogen (PDL1-Vax) are able to induce anti-PD-L1 immune responses. We found that DCs, loaded with PDL1-Vax, induced anti-PD-L1 antibody and T cell responses in immunized mice and that PD-L1-specific CTLs had cytolytic activities against PD-L1^+^ tumor cells. We demonstrated that vaccination with DCs loaded with PDL1-Vax potently inhibited the growth of PD-L1-expressing tumor cells. Thus, this study demonstrates, for the first time, the feasibility of DC vaccination (PDL1-Vax) to actively induce endogenous anti-PD-L1 antibody and T cell responses for cancer immunotherapy.

## 2. Materials and Methods

### 2.1. Tumor Cell Lines

Murine pancreatic adenocarcinoma cell line Panc02 was kindly provided by Dr. Q Yao (Baylor College of Medicine, Houston, TX, USA). Murine colon carcinoma cell line MC38 was a kind gift from Dr. John C. Morris (National Cancer Institute, Bethesda, MD, USA). These cell lines were maintained in Dulbecco’s Modified Eagle Medium (DMEM) supplemented with 10% (*v*/*v*) FBS, 1% L-glutamine, 1% sodium pyruvate, and 1% streptomycin-penicillin-neomycin solution (Biological Industries, Cromwell, CT, USA). A recombinant lentiviral PD-L1 vector that coexpresses PD-L1 and YFP (yellow fluorescent protein) was used to transduce these cells to establish PD-L1-expressing stable tumor cell lines, Panc02-PD-L1 and MC38-PD-L1. These tumor cells, which stably express human PD-L1, were maintained in the media supplemented with 500 ng/mL or 200 ng/mL puromycin (Invitrogen, Carlsbad, CA, USA).

### 2.2. Immunization of Bone Marrow (BM)-Derived Dendritic Cells (DC) and Mouse Models

Mouse BM-derived DCs were prepared, as described in our previous study [3]. The mouse BM-DCs were loaded with recombinant PDL1-Vax protein, PDL1 protein, or IgG Fc (200 µg/mL), and matured with TNF-α (200 U/mL) and PolyI:C (40 µg/mL) for additional 36–48 h. After washing with phosphate-buffered saline (PBS) three times, the BM-DCs were resuspended in PBS containing recombinant PDL1-Vax protein, PDL1 protein, or IgG Fc (200 µg/mL). The antigen-loaded BM-DCs with >75% of viability, >50% of CD11c^+^ and >40% of CD80^+^ cells were used for immunization. Six- to eight-week-old female C57BL/6 or Balb/c mice were purchased from Jackson Laboratory and maintained locally. All animal procedures were approved by the Institutional Animal Care and Use Committee at the University of Southern California. Mice were randomly divided into various groups. The mice were immunized with DCs via footpad injection, two times, at a one-week interval. At various days after DC immunization, the mice were sacrificed for monitoring anti-PD-L1 humoral and cellular immune responses. To test anti-tumor activities, the mice were subcutaneously inoculated with exponentially growing tumor cells on the flanks and, at the indicated days after tumor inoculation, immunized with antigen-loaded DCs via footpad injection, two times at a one-week interval. Tumor sizes were measured every three or four days with a caliper. Tumor volume was calculated as follows: (longest diameter) × (shortest diameter)^2^.

### 2.3. Enzyme-Linked Immunospot (Elispot) Assay

ELISPOT assay of cytokine-producing T cells was performed as described in our previous studies [33]. Briefly, Elispot plates (MilliporeSigma, St. Louis, MO, USA) were coated with monoclonal anti-IFN-γ antibody (100 ng/well) overnight followed by incubation with RPMI-1640 media supplemented with 10% FBS. T cells were isolated from splenocytes using T cell isolation kits (Miltenyi Biotec, San Diego, CA, USA) Effector T cells (1 × 10^5^, or 2 × 10^5^) were seeded into wells in the Elispot plates and cultured in the presence of antigen (50 ng per well) for 24 h at 37 °C. After extensive washing, the plates were added with biotinylated anti-mouse IFN-γ detection antibody and then incubated for 2 h. The plates were extensively washed and then added into the wells with HRP-conjugated avidin. After incubation for 1 h at room temperature, the plates were added with freshly-prepared AEC Substrate Solution (Pierce Chemical, Dallas, TX, USA) and then monitored development of spots. After spot development, the plate was rinsed thoroughly with ddH_2_O and allowed to dry. Spots were analyzed by a dissecting microscope and automated ELISPOT plate reader (Carl Zeiss, Thornwood, NY, USA).

For ELISPOT Assay of PD-L1 antibody-producing B cells, splenocytes were isolated form the immunized mice. MultiScreen-HA plates were coated with 0.5 µg of PD-L1 protein for overnight. Splenocytes (2 × 10^5^) were added to each well, and plates were incubated at 37 °C for overnight. The cells were washed away, and secreted antibodies detected with HRP-conjugated goat anti-mouse IgG (1 mg/mL) (Santa Cruz Biotechnology, Dallas, TX, USA).

### 2.4. Antibody and Cytokine ELISA

ELISA plates were coated with recombinant PD-L1 proteins (5 μg/mL) overnight at 4 °C. The PD-L1-coated plates were added with serial dilutions of sera and incubated for 1 h at room temperature. After extensive washes, the plates were added with biotinylated anti-mouse antibodies (anti-mouse IgG, IgG1, IgG2a, IgG2b, or IgG3) (Sigma-Aldrich, St. Louis, MO, USA) and incubated for 1 h at room temperature. Streptavidin-HRP was used to detect ELISA reactions. Optical densities (OD) were read at 450 nm on a BioAssay Reader (PerkinElmer, Wellesley, CA, USA). The titers are expressed as reciprocal endpoint dilutions that reach ≥2-fold OD450 values of control mouse sera.

### 2.5. PD-1/PD-L1 Inhibition Assay

96-well ELISA plates were coated with recombinant PD-L1 protein (1 μg/well). 50 μL mixture of 20 ng PD-1-biotin and sera of immunized mice, or anti-PD-L1 antibody control at indicated concentration, or 50 μL assay buffer (blank) was added into wells, and incubated at room temperature for 2 h. Diluted streptavidin-horseradish peroxidase (HRP) was added to each well after wash and incubated at room temperature for 1 h with slow shaking. After three times of wash, TMB (3,3′,5,5′-Tetramethylbenzidine) HRP substrate was added until blue color is developed in the positive control well. OD value at 450 nm UV was measured after 100 μL 2N sulfuric acid solution was added to stop reaction. The percent of inhibition activity is represented by 1 − (OD of unknown-OD of blank)/(OD of positive control-OD of blank).

### 2.6. CTL Assay

CTL assay was performed as described in our previous study [34]. Briefly, immunized mice were euthanized, and splenocytes were harvested. The splenocytes were in vitro stimulated with recombinant PD-L1 protein in 24-well plates for 6 days and used as Effector cells (E). Tumor cells were labeled with 300 μCi of ^51^Cr for 90 min and used as Target cells (T). Effector cells were mixed with Target cells at various E:T cell ratios and plated into 96-well plates in triplicate. After 4 h of incubation, supernatant was collected and counted with a gamma counter (Beckman, Fullerton, CA, USA). The cytotoxicity was calculated as follows: percentage of lysis = (sample cpm × spontaneous cpm)/(total cpm − spontaneous cpm) × 100.

### 2.7. Statistical Analysis

Statistical significance was determined using the *t*-test and One-Way or Two-Way ANOVA test with SigmaStat software. *p* < 0.05 was considered as a statistically significant difference. Regression plots were constructed using SigmaPlot software (San Jose, CA, USA). All data were presented as the mean ± SEM and were representative of at least three-independent experiments done in triplicate.

## 3. Results

### 3.1. Production of Recombinant PD-L1 Immunogens (PDL1-Vax)

Our previous studies demonstrated that linking an antigen to a DC-targeting molecule, such as IgG-Fc and heat shock protein (HSP) for receptor-mediated internalization, antigen processing, and antigen presentation, as well as DC maturation provides a means to enhance antigen-specific cellular and humoral responses for both DC and DNA vaccines [3,6,7,35,36,37,38]. To generate a PD-L1 immunogen (PDL1-Vax), a fusion gene consisting of the extracellular domain of human PD-L1 (aa 19–220) in-frame linked to a T helper epitope sequence and a human IgG1 Fc sequence was synthesized and cloned into Novagen pET28a expression vector to generate the expression vector pET-PDL1-Vax. For the expression of the recombinant protein (PD-L1-Vax), this recombinant plasmid was transformed into BL21 (*E. coli*) in the culture medium with Kanamycin. After culture fermentation and isopropyl β-D-1-thiogalactopyranoside (IPTG) induction, the cells were collected, lysed, and examined by sodium dodecyl sulfate–polyacrylamide gel electrophoresis (SDS-PAGE) (Figure 1A). The inclusion bodies containing the recombinant protein were washed extensively and then denatured with 8 M urea. The soluble proteins were subjected to anion exchange chromatography with a Source 30Q column and the recombinant protein was eluted with a buffer containing 70 mM sodium chloride. The eluted protein of interest was subjected to anion exchange chromatography with an HiTrip Q HP column to remove endotoxins and other contaminants. The eluted protein was loaded and washed with a buffer containing 1% Triton X-114, and the recombinant protein was then eluted with a buffer containing 100 mM sodium chloride. The eluted protein was subjected to Superdex 200 size exclusion column chromatography for further purification ad renaturation. The fractions of the recombinant protein were collected and the purified recombinant protein (PDL1-Vax) was further examined by SDS-PAGE with Coomassie blue staining (Figure 1B) and Western blot with an anti-PD-L1 or IgG Fc antibody (Figure 1C). Other quality control testing of endotoxins, mycoplasma and microorganisms were also performed. The recombinant protein PDL1-Vax with the purity of >90% at an acceptable low level of endotoxins were used for further studies.

### 3.2. Induction of Anti-PD-L1 Antibody Response by PDL1-Vax-Loaded DC Vaccination

We investigated whether DCs loaded with PDL1-Vax can induce PD-L1-specific antibody responses. The results of our previous studies demonstrated that DC vaccination is able to induce antibody responses [3,4,5,39,40]. It is known that human IgG Fc functions in mice [41,42]. Mouse BM-derived DCs were loaded with recombinant PDL1-Vax proteins, the PDL1 protein, or IgG Fc fragment protein, respectively, and then matured with PolyI:C ex vivo. Poly I:C, a synthetic analogue of viral dsRNA, has long been known as a strong inducer of innate immune responses against infectious diseases and cancers by recognizing Toll-like receptors (TLR) 3 in the endosome [43]. It also stimulates immune responses by recognizing the cytoplasmic dsRNA sensors, Melanoma Differentiation-Associated protein 5 (MDA-5), and Retinoic acid-inducible gene I (RIG-I) [44,45,46]. Groups of mice were immunized with the DCs loaded with various antigens twice at a weekly interval. Figure 2A shows that PDL1-Vax-loaded DCs elicited PDL1-specific IgG responses, while DCs loaded with PDL1 protein only induced weak anti-PD-L1 responses. DCs, loaded with IgG1 Fc, did not induce anti-PD-L1 responses. We also performed assays for PDL1-specific antibody subclass profiling. Figure 2B shows that PDL1-Vax-loaded DCs induced a broad responses of IgG subclasses, including IgG1 and IgG2a. Similar results were obtained in repeated experiments. To test the ability of PDL1-Vax-loaded DCs to activate PD-L1-specific B cells, we used an anti-PD-L1 IgG-specific B cell ELISPOT assay to directly examine the frequencies of anti-PD-L1 IgG-producing B cells in the immunized mice. Figure 3 shows that frequencies of anti-PD-L1 IgG-producing B cells were significantly higher in PDL1-Vax-DC-immunized mice than in PDL1-DC-immunized mice and IgG Fc-DC-immunized mice. Moreover, the sera of the mice immunized with PDL1-Vax-loaded DCs inhibited the PD-1/PD-L1 interaction, as manifested by a competitive ELISA assay, while the sera of the mice immunized with PDL1-loaded DCs only weakly inhibited the PD-1/PD-L1 interaction and the sera of the mice immunized with IgG Fc-loaded DCs did not inhibit the PD-1/PD-L1 interaction (Figure 4). Taken together, these results demonstrate that vaccinations with DCs loaded with PDL1-Vax induce anti-PD-L1 antibody responses.

### 3.3. Induction of PD-L1-Specific T Cell Response by PDL1-Vax DC Vaccination

We investigated whether immunization with PDL1-Vax-DCs can induce PD-L1-specific T cell responses. Groups of mice were immunized with DCs loaded with PDL1-Vax, PDL1 or IgG Fc twice at a weekly interval. Two weeks later, CD3^+^ T cells were isolated from the splenocytes of immunized mice for ELISPOT assays [3,4,5,39]. Figure 5A shows that DCs loaded with PDL1-Vax induced stronger CD3^+^ T cell response than DCs loaded with PDL1 or IgG Fc. We further isolated the CD3^+^CD8^+^ CTL cells for ELISPOT assays. Consistent with the results of total CD3^+^ T cells, DCs loaded with PDL1-Vax were more potent than DCs loaded with PDL1-Vax in inducing PD-L1-specific CD8^+^ CTL responses (Figure 5B). We also determined whether immunization with transduced DCs can induce CD4^+^ Th responses. Figure 5C shows the ability of PDL1-Vax-DCs to induce CD4^+^ Th responses as well.

To determine whether immunization with PDL1-Vax DCs can indeed induce functional CTL responses, we performed the CTL assays [3,4,5,39]. Splenocytes from immunized mice in different groups were restimulated with the recombinant PD-L1 protein ex vivo and then cocultured with ^51^Cr-labeled syngeneic target PD-L1-expressing pancreatic cancer cells (Panc02-PD-L1) at various E:T ratios to measure the specific killing. Splenocytes from the mice immunized with PDL1-Vax-DCs killed target cells much more efficiently than those from mice immunized with PDL1-DCs or IgG Fc-DCs (Figure 6). Thus, these results demonstrate the ability of PDL1-Vax-DCs to induce PD-L1-specific CTL responses.

### 3.4. Inhibition of PD-L1+ Tumor Growth by PDL1-Vax DC Vaccination

We further examined the ability of PDL1-Vax DC vaccination to inhibit PD-L1^+^ tumor growth in mice. Panc02-PD-L1 pancreatic tumor cells were inoculated subcutaneously in the flank of syngeneic C57BL/6 mice. Three days later, the mice were randomly assigned into different groups and then administered s.c. with DCs loaded with PDL1-Vax, PDL1, or IgG Fc twice at a weekly interval. Tumor growth of each group of mice was then monitored. As shown in Figure 7, PDL1-Vax-DC vaccinations significantly inhibited the growth of Panc02-PD-L1 tumor. PDL1-DC vaccinations only weakly inhibited the growth of Panc02-PD-L1 tumor and IgG Fc-DC vaccinations did not inhibit the growth of Panc02-PD-L1 tumor. Repeated experiments showed similar results. We also examined the ability of PDL1-Vax-DC vaccination to inhibit the growth of other types of PD-L1^+^ tumor cells. Consistently, PDL1-Vax-DC vaccinations also significantly inhibited the growth of murine colon adenocarcinoma cells MC38-PD-L1 (Figure 7). Taken together, these results demonstrate that PDL1-Vax-DC vaccinations are able to inhibit the growth of PD-L1 + tumor cells.

Furthermore, we examined possible adverse autoimmune pathology induced by immunization with PDL1-Vax-DCs. No other apparent toxicity was observed in mice immunized with PDL1-Vax-DCs. Primary histological analysis of vital organs and tissues in the mice immunized with PDL1-Vax-DCs did not observe any significant pathologic inflammations, suggesting that PDL1-Vax-DC does not cause pathological toxicity in immunized mice (data not shown).

## 4. Discussion

This study demonstrates, for the first time, the principle and feasibility of vaccination with PDL1-Vax DCs to induce the production of anti-PD-L1 antibodies and to activate PD-L1-specific CTL responses capable of inhibiting PD-L1^+^ tumor growth. Different from the repeated administrations of a large amount of manufactured PD-1 or PD-L1 antibodies into cancer patients at a 2–3-week interval, this novel PD-L1 vaccination strategy provides a novel strategy to persistently produce anti-PD-L1 antibodies and T cell response for cancer treatment and prevention.

Most immune responses require the development of a balance between Th1 and Th2 subsets, since protective immunity to viruses needs the input from CTLs as well as neutralizing antibodies [47]. Although, DCs stimulating an antibody response is believed as a consequence of CD4^+^ Th function, recent evidence supports that DCs can directly induce antibody responses [48,49,50,51]. DCs promote proliferation and antibody production of CD40-activated naive and memory B cells during stimulation of B cell responses. Antigen-loaded DCs can trigger an antibody response [52]. DCs produce factors, such as IL-10 and IL-6, and express membrane/soluble proteins to promote B cell growth and differentiation [53,54,55]. DCs also produce IL-12 to boost Th1 development and promote B cells to develop humoral responses [47,56]. Our previous studies found that DCs can activate both CTL and antibody responses [3,5,39,40]. In this study, we found that DCs, loaded with a recombinant PD-L1-IgG Fc immunogen, induced both anti-PD-L1 antibody and T cell responses in immunized mice. We demonstrated that, vaccination with DCs loaded with the PD-L1 immunogen (PDL1-Vax), potently inhibited the growth of PD-L1-expressing tumor cells. The induced anti-PD-L1 antibodies can kill PD-L1-expressing tumor cells by multiple mechanisms, including the inhibition of PD-L1 interaction with PD-1 on CTLs, the antibody-dependent cellular cytotoxicity (ADCC) and *complement*-*dependent cytotoxicity* (CDC).

It was previously observed that PD-L1-specific CTLs naturally exist in cancer patients and are able to recognize and kill malignant PD-L1^+^ tumor cells [57,58,59]. Moreover, anti-PD-L1 antibodies were also detected in patients [60]. In this study, we found that DCs loaded with PDL1-Vax are more potent than DCs loaded with PD-L1 protein in inducing anti-PD-L1 immune responses and inhibiting tumor growth. The results of our study indicate that PD-L1 protein is weakly immunogenic, and the recombinant fusion protein PDL1-Vax is more immunogenic, probably due to the enhanced antigen presentation of PDL1-Vax by DCs [3,6,7]. PDL1-Vax DC vaccines can be used in combination with radiotherapy, chemotherapy and targeted therapy [61,62,63].

The immune checkpoint blockade has robust antitumor activities in some cancer patients. Unfortunately, the majority of patients did not respond or only had a partial response to PD-L1 or PD-1 inhibitors, which may be limited by the absence of sufficient anti-tumor T cells in cancer patients. In this study, we found that PDL1-Vax DCs induced anti-PD-L1 antibody T cell responses in immunized mice. The activated PD-L1-specific CTLs have the ability to kill PD-L1^+^ tumor cells. Thus, in addition to the anti-PD-L1 antibody-mediated tumor killing, the activated PD-L1-specific CTLs can additionally kill PD-L1^+^ tumor cells in the immunized mice or patients. It is important to test the novel human PDL1-Vax DC vaccine in cancer patients to determine whether anti-PD-L1 antibody and CTL responses can be induced and whether PDL1-Vax DC vaccine is safe. Although, there are no apparent signs of toxicities and inflammations observed in the immunized mice, we intend to further investigate the potential autoimmune responses and toxicities induced by the PDL1-Vax DCs in immunized mice and cancer patients.

In a recent study, Mascaux C, et al. found that immune activation and immune escape occur before tumor invasion [64]. They found the activation and infiltration of T cells and immune escape through the upregulation of PD-L1 and other immune suppressive molecule expression in the earliest pre-cancerous and pre-invasive cancerous lesions. Thus, the PDL1-Vax vaccination with the ability to induce persistent anti-PD-L1 antibody and CTL responses may be particularly suitable for immune prevention of carcinogenesis in healthy individuals at high risk of developing cancer, such as, smokers and individuals with genetic defects or chronic viral infections. Moreover, the PDL1-Vax vaccination could be also well-suited for preventing cancer recurrence after surgery and other standard treatments.

## 5. Conclusions

In this study, we designed and tested a novel anti-PD-L1 vaccine in mice. We found that DCs, loaded with a PD-L1 immunogen (PDL1-Vax), induced anti-PD-L1 antibody and T cell responses in immunized mice and that vaccination with PDL1-Vax DCs potently inhibited the growth of PD-L1^+^ tumor cells. This study demonstrates, for the first time, the principle and feasibility of DC vaccination (PDL1-Vax) in actively inducing anti-PD-L1 antibody and T cell responses, capable of inhibiting PD-L1^+^ tumor growth. This novel anti-PD-L1 vaccination strategy could be used for cancer immunotherapy and immunoprevention.

## Figures and Tables

**Figure 1 cancers-11-01909-f001:**
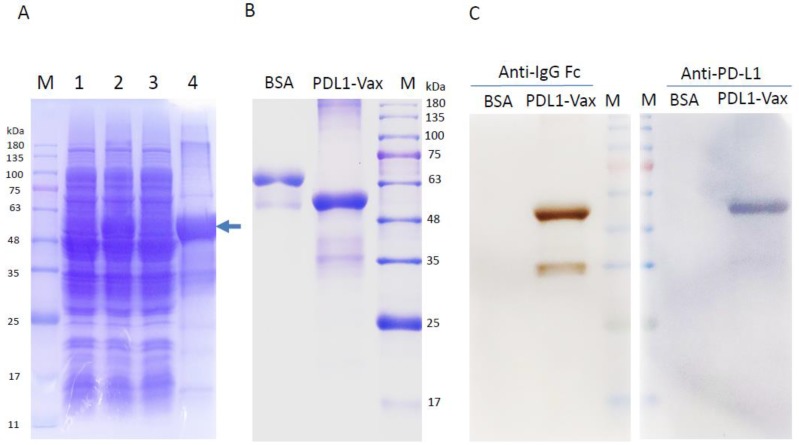
Expression and purification of the recombinant protein PDL1-Vax. (**A**) Protein fractions after lysis of bacterial cells (pET-PDL1-vax) were electrophoresed on 12% SDS–PAGE gels and then stained with Coomassie brilliant blue R250. Lane M, prestained protein molecular weight marker; lane 1, the cell pellets before IPTG induction; lanes 2, the cell pellets after isopropyl β-D-1-thiogalactopyranoside (IPTG) induction; lane 3, the soluble fraction after IPTG induction; and lanes 4, the insoluble protein fraction after IPTG induction. The recombinant protein PD-L1-Vax (arrow) is indicated. (**B**) Purified recombinant proteins PDL1-Vax were electrophoresed on 12% SDS–PAGE gels and then stained with Coomassie brilliant blue R250. (**C**) Western blot analysis of the purified recombinant protein PDL1-Vax using antibodies against human PD-L1 and IgG Fc. An irrelevant protein bovine serum albumin (BSA) was used as a negative control.

**Figure 2 cancers-11-01909-f002:**
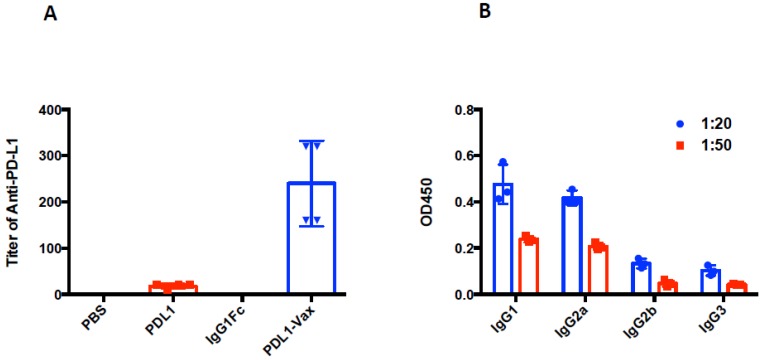
Induction of anti-PD-L1 antibody responses. Groups of C57BL/6 mice were immunized with various antigen-loaded BM-derived DCs (1 × 10^6^ cells/mouse) twice at a weekly interval, and sera were collected from each group of mice 14 d later. (**A**) PD-L1-specific IgG levels from the pooled sera of each group (4–6 mice/group) were determined by ELISA. (**B**) PD-L1-specific IgG subclass levels from the pooled sera of PDL1-Vax-DCs-immunized mice (4–6 mice/group) were also determined by ELISA. *p* < 0.01, PDL1-Vax-DCs versus PDL1-DCs or IgG Fc-DCs.

**Figure 3 cancers-11-01909-f003:**
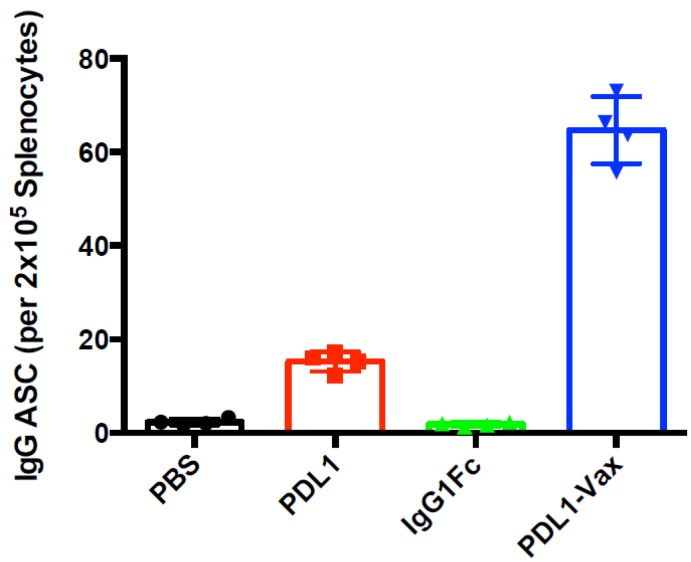
Activation of PD-L1-specific B cells. Groups of C57BL/6 mice were immunized with various antigen-loaded BM-derived DCs (1 × 10^6^ cells/mouse) twice at a weekly interval, and splenocytes were prepared from each group of mice (5 per group) 14 d later. Frequencies of anti-PD-L1 antibody-secreting B cells (ASC) in different groups of mice were determined and presented as the number of cells secreting PD-L1-specific IgG per 2 × 10^5^ B cells. *p* < 0.01, PDL1-Vax-DCs versus PDL1-DCs or IgG Fc-DCs.

**Figure 4 cancers-11-01909-f004:**
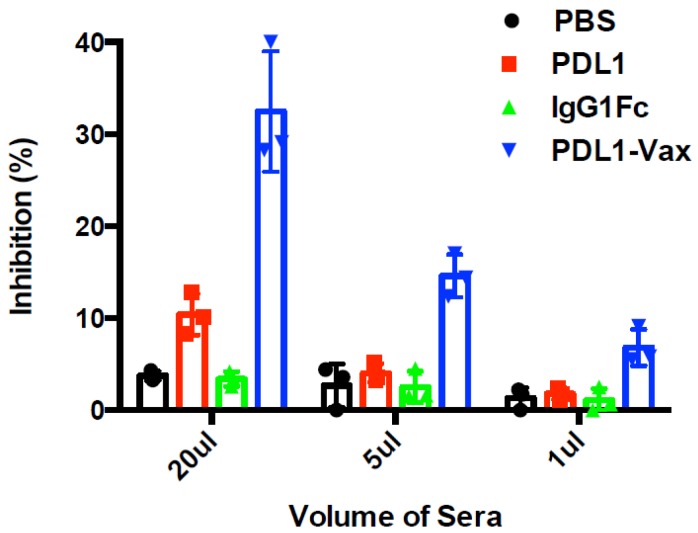
Inhibition of PD-1 and PD-L1 interaction. Sera were collected from each group of mice (immunized with various antigen-loaded BM-DCs. Inhibition of PD-1 and PD-L1 interaction by the addition of different amounts of the sera of the mice (5 per group), immunized with various antigen-loaded DCs, was performed using a competitive ELISA. The percentages of inhibition were determined and presented. *p* < 0.01, PDL1-Vax-DCs versus PDL1-DCs or IgG Fc-DCs.

**Figure 5 cancers-11-01909-f005:**
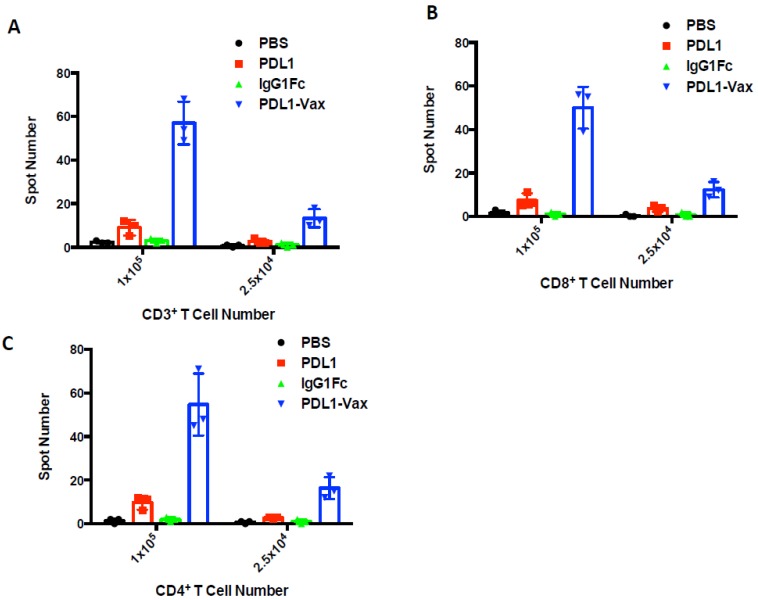
Induction of PD-L1-specific CD4^+^ Th and CD8^+^ CTL responses. Groups of C57BL/6 mice were immunized with various antigen-loaded BM-derived DCs (1 × 10^6^ cells/mouse) twice at a weekly interval, and, 14 d later, CD3^+^ T cells, CD4^+^ T cells, and CD8^+^ T cells were isolated from splenocytes of immunized mice (5 per group) were used for IFNγ ELISPOT assays. (**A**) Frequencies of PD-L1-specific CD3^+^ T cells are presented as the number of IFNγ-producing cells per 1 × 10^5^ or 2.5 × 10^4^ splenocytes. (**B**) Frequencies of PD-L1-specific CD4^+^ T cells. (**C**) Frequencies of PD-L1-specific CD8^+^ T cells. *p* < 0.01, PDL1-Vax-DCs versus PDL1-DCs or IgG Fc-DCs.

**Figure 6 cancers-11-01909-f006:**
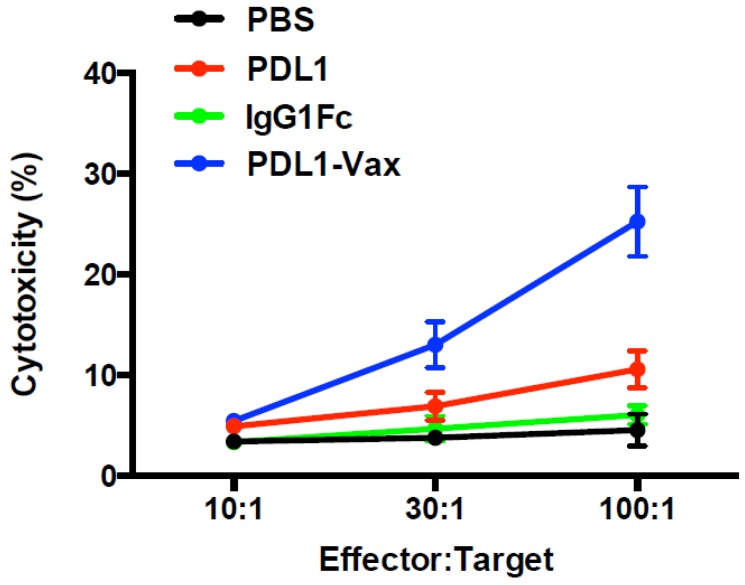
CTL assays. Groups of C57BL/6 mice were immunized with various antigen-loaded BM-derived DCs (1 × 10^6^ cells/mouse) twice at a weekly interval, and, 14 d later, splenocytes taken from different group of mice were restimulated in vitro with recombinant PD-L1 protein. The restimulated splenocytes (*E*) were cocultured with ^51^Cr-labeled target cells Panc02-PD-L1 (T) at various ratios. The percentages of target cell killing by the splenocytes from different immunized mice were determined and shown. *p* < 0.01, PDL1-Vax-DCs versus PDL1-DCs or IgG Fc-DCs. The data represent the means of triplicate samples from one representative experiment of three (six mice/group); *bars,* SE.

**Figure 7 cancers-11-01909-f007:**
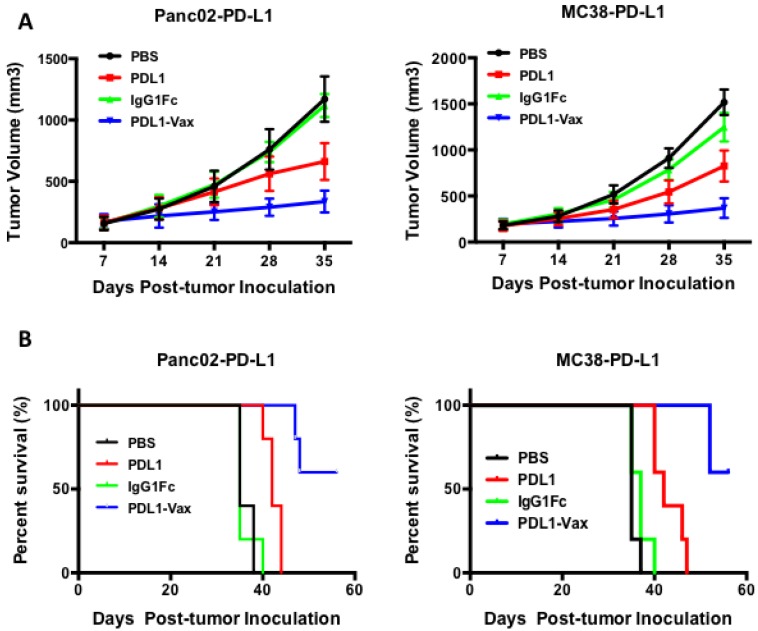
Induction of antitumor immunity. Groups of mice (5 per group) were inoculated with 3 × 10^5^ exponentially growing PD-L1^+^ tumor cells (murine pancreatic adenocarcinoma cells Panc02-PD-L1 or murine colon carcinoma cells MC38-PD-L1). Seven days later, mice were immunized with various antigen-loaded BM-derived DCs (1 × 10^6^ cells/mouse) twice at a weekly interval. Tumor sizes were measured every 3–4 days. (**A**). Tumor volumes in each group are presented. *p* < 0.01, PDL1-Vax-DCs versus PDL1-DCs or IgG Fc-DCs. Data represent the means of one of three independent experiments; *bars*, SE. (**B**). Mouse death was recorded for evaluation of mouse survival percentage.

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
