# Peer review of "A Novel Anti-PD-L1 Vaccine for Cancer Immunotherapy and Immunoprevention"

_cancers, 2019, doi:10.3390/cancers11121909_

Round 1

Reviewer 1 Report

OK for me now

Reviewer 2 Report

The revision has satisfactorily addressed my concern and this manuscript is now acceptable for publication.

Reviewer 3 Report

The authors have appropriately addressed the comments from last time. 

This manuscript is a resubmission of an earlier submission. The following is a list of the peer review reports and author responses from that submission.

Round 1

Reviewer 1 Report

In this in this interesting manuscript the authors provide a method to produce a DC-based vaccine against the immune checkpoint protein PD-L1. The study is well conducted, the data are convincing and supporting the authors conclusions. However, I have only major conceptual concern. It is well documented that some cancers overexpress PD-L1, helping them to escape immunesurveillance, thus, PD-L1 can be a suitable tumor antigen towards which mounting both humoral and cellular immune response. However, the major known function of immune checkpoint is to keep the immune system cool towards self and weak antigens. Thus, the recent success of anti immune checkpoint therapy with monoclonal antibody is such also because it is time limited, until immune-mediated rejection of the tumor. This helps reducing the risk of serious autoimmune and inflammatory collateral effects. A steady immunization against PD-L1, as suggested by the authors, might perhaps combat cancer but with the risk of creating a bad quality of following life by autoimmune disease. The authors briefly mention that mice do not show signs of inflamed organs or tissue upon immunization, however, they do not show data nor indicate how long mice have been followed up for.  Of course, this type of potential collateral problem would be very difficult to predict in humans even from evidence in mice. 

Author Response

Response:

Thank Reviewer #1 for the insightful comments, As a result of the reviewer’s concern on the potential toxicities, we have revised the text of the MS to include the following description:

“Although there are no apparent signs of toxicities and inflammations observed in the immunized mice, we intend to further investigate the potential autoimmune responses and toxicities induced by the PDL1-Vax DCs in immunized mice and cancer patients” (P. 10, underlined).

Reviewer 2 Report

In this communication, Chen et al report that a chimeric molecule composing of a portion of PD-L1 and a human IgG1 Fc could be efficiently loaded to dendritic cells (DCs) for efficient induction of both humoral and cellular anti-PD-L1 immunity. Their data further show that the combined effect of the induced humoral and cellular immunity could efficiently inhibit the growth of cells harboring PD-L1 in two separate tumor models. Overall, this article is well written and the data support the conclusion. The unique construct (PDL1-Vax) provides substantial novelty and the obtained results have implications for clinical translation.

Minor comments

Please fix this sentence and add the citation following the sentence “Recently, we found that a genetically modified DC vaccine elicited potent tumor-47 associated antigen-specific T cell responses yielded improved survival rate in acute leukemia patients.” The human PD-L1 was used in this study and it was presented as “foreign” in both animal models. The authors need to discuss the potential difficult in generating robust anti-PD-L1 immunity in human patients where this molecule is not “foreign” anymore.

Author Response

Response:

In response to Reviewer #2’s concerns, the following revisions are made:

A citation is added (11) and the sentence is revised:  “Recently, we found that a genetically modified DC vaccine elicited potent tumor associated antigen-specific T cell responses and yielded improved survival rate in acute leukemia patients (11).” (P. 2). A sentence is added in the Discussion.  "It is important to test the novel human PDL1-Vax DC vaccine in cancer patients to determine whether anti-PD-L1 antibody and CTL responses can be induced and whether PDL1-Vax DC vaccine is safe.” (P.10, underlined).

Reviewer 3 Report

Overall, the article is very well written and the content is novel. 

I just a few minor concerns and one major concern: 

Minor concerns: 

In experiments involving mice, please include sample size in text and figure legend. 

For the experiment where the PDL1-Vax was tested in-vivo in mouse models of MC-38 and Panc02, please include data showing weight of excised tumors at the end of the study. As variations is subq tumor volumes are sometimes due to differences in how deep or superficial the tumors were implanted, tumor weights will help rule out any concerns. 

Was there any difference in survival in the above experiment? Add survival curves. 

Toxicities or immune related adverse events with administration of PD-1 and PDL-1 antibodies is a cause for concern in the clinic. While it is extremely interesting to see improvement in anti-tumor response with PDL-1 Vax, one cannot avoid addressing potential toxicities with PDL-1 Vax. Though the toxicities were tested, the data is not shown. I think it is really important to include the histology data. 

Major concern: 

The study is funded by Pomona and Jiyan Biotech. Was the PDL-1 Vax provided by the company/companies. I believe Pomona has iPDL-1 Vax commercially available. Is it the same drug? Is there a conflict of interest? Please address any conflicts of interest. 

Author Response

Response:

In response to Reviewer #3’s minor concerns, the following revisions are made:

Mouse survival curves are provided in the revised manuscript (Fig. 7B). A sentence is added in the Discussion section: “Although there are no apparent signs of toxicities and inflammations observed in the immunized mice, we intend to further investigate the potential autoimmune responses and toxicities induced by the PDL1-Vax DCs in immunized mice and cancer patients” (P. 10, underlined).

Response to the Reviewer’s Major concern: 

This manuscript reports a mouse study that demonstrate the design and function of a novel PDL1-Vax DC vaccine in mice.  The conflict of interest information is provided in the revised manuscript.